# Pneumothorax after CT-guided transthoracic lung biopsy: A comparison between immediate and delayed occurrence

**Kyungsoo Bae**[1,2], **Ji Young Ha**[1,2], **Kyung Nyeo Jeon**[1,2]*

**1** Department of Radiology, Institute of Health Sciences, Gyeongsang National University School of Medicine, Jinju, Korea, **2** Department of Radiology, Gyeongsang National University Changwon Hospital, Changwon, Korea

* knjeon@gnu.ac.kr

## Abstract

### Background

In CT-guided transthoracic lung biopsy (CTLB), pneumothorax can occur as a late complication (delayed pneumothorax). The incidence, risk factors, and clinical significance of delayed pneumothorax are not well known.

### Objectives

To compare the risk factors for immediate and delayed pneumothorax after CTLB and to know their clinical significance.

### Methods

Images and medical records of 536 consecutive patients who underwent CTLB were reviewed. All biopsies were performed as inpatient procedures. Follow-up chest radiographs were obtained at least twice at 4 h after procedure and before discharge. Risk factors for immediate and delayed pneumothorax were assessed based on patient-, lesion-, and procedure-related variables. Rates of chest tube insertion were also compared.

### Results

Pneumothorax developed in 161 patients (30.0%) including 135 (25.2%) immediate and 26 (4.9%) delayed cases. Lesion size was an independent risk factor for both immediate and delayed pneumothorax (OR = 0.813; CI = 0.717–0.922 and OR = 0.610; CI = 0.441–0.844, respectively). While emphysema, lower lobe location, and long intrapulmonary biopsy track were risk factors (OR = 1.981; CI = 1.172–3.344, OR = 3.505; CI = 2.718–5.650, and OR = 1.330; CI = 1.132–1.563, respectively) for immediate pneumothorax, upper lobe location and increased number of pleural punctures were independent risk factors (OR = 5.756; CI = 1.634–20.274 and OR = 3.738; CI = 1.860–7.511, respectively) for delayed pneumothorax. The rate of chest tube insertion was significantly ($p < 0.001$) higher in delayed pneumothorax.

**Data Availability Statement:** All relevant data are within the paper and its Supporting Information file.

**Funding:** The authors received no specific funding for this work.

**Competing interests:** No authors have competing interests.

**Abbreviations:** CTLB, CT-guided transthoracic lung biopsy; CT, Computed tomography; MDCT, Multi-detector computed tomography; GGO, Ground-glass opacity; OR, Odds ratio; CI, Confidence interval.

## Conclusion

Pneumothorax tends to occur immediately after CTLB in patients with emphysema, lower lobe lesion, and long intrapulmonary biopsy track. Further attention and warnings are needed for those with multiple punctures of small lesions involving upper lobes due to the possibility of delayed development of pneumothorax and higher requirement for chest tube drainage.

## Introduction

The demand for lung biopsy is growing due to an increase in detection of lung tumor in screening and the need for molecular and genomic profiling of non-small cell lung cancer [1–3]. Among various imaging tools including CT, fluoroscopy, and ultrasound, CT is the preferred guidance method for transthoracic lung biopsies due to its high spatial and contrast resolution [4–6]. Pneumothorax is one of most frequent and potentially dangerous complication associated with this procedure [7]. A timely diagnosis of pneumothorax is clinically important for management of patients.

Risk factors for pneumothorax after transthoracic lung biopsy have been examined widely. Among the known risk factors, some are consistent whereas others are inconsistent or even contradictory across published studies, probably owing to various baseline characteristics, biopsy techniques, and analytic methods [8–12]. Although biopsy related pneumothorax frequently occurs during or immediately after procedure, pneumothorax can be identified in the follow-up chest radiographs or even after discharge due to chest pain or dyspnea (delayed pneumothorax) [13, 14]. In many institutions, biopsy is performed as an outpatient procedure with early discharge [15, 16]. Effective monitoring of patients after procedure requires detection of patients who are at risk of delayed development of pneumothorax. Most previous studies regarding pneumothorax have focused on incidence rates and overall risk factors. Only a few studies have given attention to the developing time of pneumothorax [17–19]. Therefore, the incidence, risk factors, and clinical significance of delayed pneumothorax are not well known.

Since CTLBs were performed in inpatient setting at our institution, the precise rate of delayed pneumothorax could be obtained. The purpose of this study was to determine the risk factors and clinical significance of delayed pneumothorax after CTLB, compared to those of immediate pneumothorax.

## Methods

This study was approved by Institutional Review Board of Gyeongsang National University Changwon Hospital. The requirements for written informed consent were waived owing to the retrospective nature of the study.

### Study population

Between March 2010 and September 2015, 578 consecutive patients underwent CT-guided transthoracic biopsy conducted by two thoracic radiologists (with 16 and 7 years of experience in image-guided biopsy). After exclusion of 42 patients (28 with pleural or mediastinal lesions, nine with multiple lung biopsy during the same admission period, three with ipsilateral chest tube insertion, and two cases of technical failure), 536 patients who underwent lung biopsy were enrolled. There were 372 males and 164 females with a mean age of 65.8 (range, 18–90) years. In

all patients, the platelet count exceeded 100,000/ μL, and the prothrombin time and activated partial prothrombin time were within normal limits. CTLB was performed in the inpatient setting. Written informed consent was obtained from all patients before undergoing procedures.

## Biopsy procedure and chest radiographic follow-up

Procedures were performed under the guidance of 16-slice MDCT scanner (LightSpeed 16, GE Healthcare). An 18-gauge core needle (Bard Magnum, Covington) with an automated biopsy gun was used in all cases. The procedures were performed with patients in prone, supine, oblique, or lateral decubitus positions depending on lesion location. After puncturing the skin, the patients were instructed to hold their breath at a normal expiration and the pleural puncture was subsequently made. The position of the needle tip was confirmed by obtaining limited CT images of 3–5 mm thickness around the lesion. After completion of tissue sampling, all patients underwent immediate CT scanning on the table to detect procedure-related complications. Patients were asked to lie with the puncture site down and coughing and ambulation was discouraged for the first 4 h. Inspiration upright posteroanterior chest radiograph was routinely performed at 4 h after procedure and prior to discharge. However, depending on patients' condition, the examination time was modified or additional chest radiographs were obtained. If no major complications were detected, patients were discharged within 24 h after admission. A drainage catheter or chest tube was inserted in symptomatic patients when displaced lateral visceral pleural line was visible on chest radiographs. An overall flow diagram of the methodology undertaken in the study is presented in Fig 1.

## Data collection

Pneumothorax was considered to be "immediate" if it was detected during the procedure or in immediate post-biopsy chest CT scan. Pneumothorax was considered to be "delayed" when it was first detected in follow-up chest radiographs after biopsy.

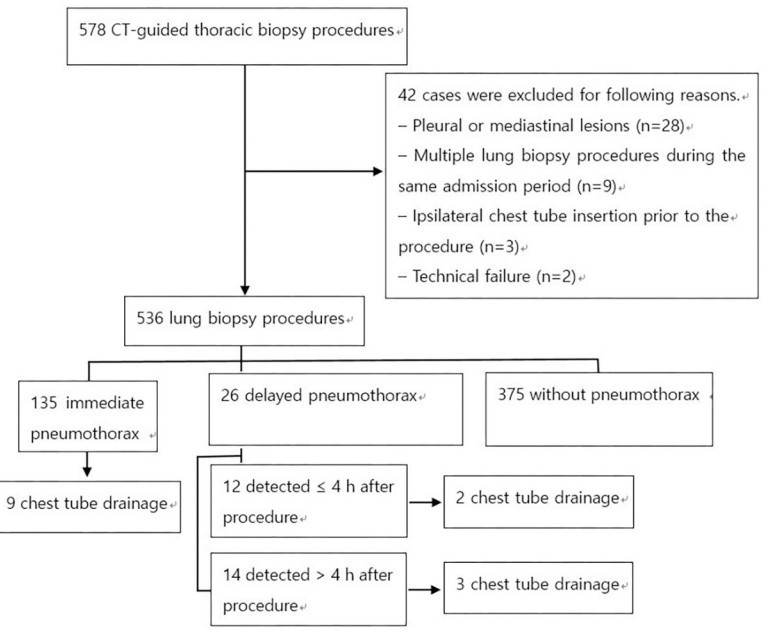

**Fig 1. Flow diagram of the study.**

Patient's age and gender, lesion size, lesion location (upper lobe vs. middle/lower lobe), lesion type (solid nodule/mass vs. consolidation/GGO), pathologic results (benign vs. malignancy), number of pleural punctures, pleural puncture angle (vertical vs. oblique), length of intrapulmonary biopsy track, needle indwelling time, and the presence or absence of emphysema in the affected lobe were compared among different groups (no pneumothorax, total pneumothorax, immediate pneumothorax, and delayed pneumothorax). The rates of chest tube insertion between immediate and delayed pneumothorax were compared. Patients' demographics, lesion characteristics, and procedure-related variables are summarized in Table 1.

## Statistical analysis

Factors related to patients, lesions, and procedures were compared via bivariate analyses using the two-sided Student's t-test or Mann-Whitney U test for numeric values and chi-square test for categorical values. The factors that were significant in the bivariate analyses were used as variables in multivariate logistic regression to identify independent risk factors for pneumothorax. Odds ratios (ORs) with 95% CIs were calculated. Differences were considered significant at $p < 0.05$. Statistical analyses were performed using commercial software (SPSS, version 24.0, SPSS Inc.).

## Results

Pneumothorax developed in 161 patients (30.0%). They include 135 cases (83.9%) of immediate pneumothorax and 26 cases (16.1%) of delayed pneumothorax. The initial follow-up chest radiograph taken at 4 h after procedure revealed persistence of immediate pneumothorax in 55 of 135 cases (40.7%). 12 cases (12/26, 46.2%) of delayed pneumothorax were detected on the initial

**Table 1. Demographics, lesion characteristics, procedural factors, and incidence of chest tube insertion.**

| Variables | All patients (n = 536) | No PNX (n = 375) | Total PNX (n = 161) | Immediate PNX (n = 135) | Delayed PNX (n = 26) |
|---|---|---|---|---|---|
| Age, yr (range) | 65.8 (18–90) | 65.4 (18–86) | 66.8 (31–90) | 66.9 (31–90) | 66.5 (50–88) |
| Male: female | 372:164 | 254: 121 | 118: 43 | 100: 35 | 18: 8 |
| Lesion size, cm (range) | 4.34 ± 2.19 (0.7–14) | 4.63 ± 2.29 (0.7–14) | 3.68 ± 2.29 † (1.0–10.8) | 3.76 ± 1.83 ‡ (1.4–10.8) | 3.26 ± 1.44 § (1.0–7.3) |
| Lesion location, upper: middle/lower | 281:255 | 214:161 | 67:94 † | 46:89 ‡ | 21:5 § |
| Lesion type, solid: lepidic | 473: 63 | 324: 51 | 149: 12 | 123: 12 | 26: 0 |
| Emphysema in affected lobe, n (%) | 205 (38.35%) | 129 (34.40%) | 76 (47.20%) † | 65 (48.15%) ‡ | 11 (42.31%) |
| Final diagnosis, malignancy: benign | 337: 199 | 231: 144 | 106: 55 | 88: 47 | 18: 8 |
| No. pleural puncture, mean (range) | 1.38 ± 0.71 (1–4) | 1.33 ± 0.69 (1–4) | 1.48 ± 0.75 † (1–4) | 1.41 ± 0.71 (1–4) | 1.85 ± 0.83 § (1–3) |
| Pleural puncture angle, vertical: oblique | 432: 104 | 307:68 | 125: 36 | 102: 33 | 23: 3 |
| Intrapulmonary biopsy track, cm (range) | 1.55 ± 1.45 (0.1–7.0) | 1.43 ± 1.43 (0.1–6.7) | 1.85 ± 1.44 † (0.1–7.0) | 1.80 ± 1.48 ‡ (0.1–7.0) | 2.15 ± 1.22 § (0.2–4.8) |
| Needle indwelling time, sec (range) | 142.94 ± 57. 54 (55–585) | 143.19 ± 59.61 (64–585) | 142.37 ± 52.58 (55–486) | 143.34 ± 52.18 (55–486) | 137.35 ± 55.37 (84–333) |
| Chest tube insertion, N (%) | 14 (2.61%) | 0 | 14 (8.70%) | 9 (6.67%) | 5 (19.23%) ** |

PNX, pneumothorax

Data are presented as mean ± SD or No (range) unless otherwise specified.

†Significant difference compared with patients without pneumothorax.

‡Significant difference compared with patients without pneumothorax.

§Significant difference compared with patients without pneumothorax.

**Significant difference compared with patients with immediate pneumothorax.

follow-up chest radiograph. 14 cases was identified more than 4 h after biopsy. The cumulative incidence of pneumothorax according to the time of detection is presented in Table 2.

In univariate analysis, the pneumothorax group had smaller lesions ($p < 0.001$), middle/lower lobe location ($p = 0.001$), longer intrapulmonary needle track ($p = 0.002$), increased number of pleural punctures ($p = 0.02$), and emphysema ($p = 0.009$), compared to the group without pneumothorax. Immediate pneumothorax was associated with smaller lesions ($p < 0.001$), middle/lower lobe location ($p < 0.001$), longer intrapulmonary needle track ($p = 0.01$), and emphysema ($p = 0.007$), compared to the group without pneumothorax. Delayed pneumothorax group had smaller lesion ($p < 0.001$), upper lobe location ($p = 0.02$), and increased number of pleural punctures ($p < 0.001$), compared to the group without pneumothorax (Table 1). Age, gender, type of lesion, pleural puncture angle, pathological results, and needle indwelling time did not show any differences between groups with and without pneumothorax.

In multivariate analysis, lesion size (odds ratio [OR] = 0.779; 95% confidence interval [CI] = 0.690–0.878), middle/lower lobe location (OR = 2.344; CI = 1.524–3.610), long intrapulmonary biopsy track (OR = 1.25; CI = 1.139–1.541), increased number of pleural punctures (OR = 1.604; CI = 1.153–2.235), and presence of emphysema in affected lobe (OR = 2.042; CI = 1.255–3.322) were risk factors for pneumothorax (Table 3). Risk factors for immediate pneumothorax were lesion size (OR = 0.813; CI = 0.717–0.922), middle/lower lobe location (OR = 3.505; CI = 2.718–5.650), long intrapulmonary biopsy track (OR = 1.330; CI = 1.132–1.563), and emphysema (OR = 1.981; CI = 1.172–3.344) (Table 3). Risk factors for delayed pneumothorax were lesion size (OR = 0.610; CI = 0.441–0.844), upper lobe location (OR = 5.756; CI = 1.634–20.274), and increased number of pleural punctures (OR = 3.738; CI = 1.860–7.511) (Tables 3 and S1). Emphysema and long intrapulmonary needle track were not risk factors for delayed pneumothorax. The intrapulmonary needle track was significantly longer in the upper lobes than in the middle/lower lobes ($1.81 \pm 1.56$ cm vs. $1.27 \pm 1.25$ cm, $p < 0.001$) in delayed group.

Fourteen of 161 cases (8.7%) of pneumothorax required chest tube or drainage catheter placement. Among them, 9 cases were in immediate group and 5 cases were in delayed group. The rate of chest tube insertion was significantly higher in delayed group (19.2%) than in immediate group (6.7%) ($P < 0.001$). Among 9 cases of immediate pneumothorax requiring chest tube, tube insertion was conducted immediately after biopsy procedure in 2 cases. In 7 cases, tube insertion was conducted after the initial 4 h follow-up chest radiographs. When comparing chest tube insertion rates between immediate pneumothorax that was persistently shown on initial follow-up chest radiographs (7/55, 12.7%) and delayed pneumothorax (5/26, 19.2%), the difference was not significant.

**Table 2. The cumulative incidence of pneumothorax (PNX) according to the time of detection (n = 161).**

| Time | Number of patients (cumulative %) |
| --- | --- |
| 0 (Immediate PNX) | 135 (83.9%) |
| $\leq$ 4 h | 12 (91.3%) |
| $\leq$ 8h | 3 (93.2%) |
| $\leq$ 16h | 6 (96.9%) |
| $\leq$ 24 h | 3 (98.8%) |
| > 24 | 2 (100%) |

PNX, pneumothorax

**Table 3. Multivariate analysis of risk factors for development of pneumothorax.**

| | Total PNX | Immediate PNX | Delayed PNX |
|---|---|---|---|
| **Lesion size (cm)** | 0.779 (CI = 0.690–0.878) | 0.813 (CI = 0.717–0.922) | 0.610 (CI = 0.441–0.844) |
| **Middle/lower lobe: Upper lobe** | 2.344 (CI = 1.524–3.610) | 3.505 (CI = 2.178–5.650) | 0.174 (CI = 0.049–0.612) |
| **Emphysema in affected lobe** | 2.042 (CI = 1.255–3.322) | 1.981 (CI = 1.172–3.344) | |
| **No. of pleural punctures** | 1.604 (CI = 1.153–2.235) | | 3.738 (CI = 1.860–7.511) |
| **Intrapulmonary biopsy track (cm)** | 1.325 (CI = 1.139–1.541) | 1.330 (CI = 1.132–1.563) | |

PNX, pneumothorax

Data are presented as odds ratios (95% confidence interval [CI]).

No pneumothorax group was used as the reference.

## Discussion

In the present study, pneumothorax developed in 30% of CTLB procedures, comparable to previous studies [9, 20]. Of all cases of pneumothorax, 16.1% was delayed occurrence. In other studies involving delayed pneumothorax, the proportion ranged from 7.1% to 29.6% of the overall rate of pneumothorax [17–21].

Among variables contributing to the risk of pneumothorax, factors other than small lesion size remain controversial [9, 19, 21–24]. Our study showed small lesion size as the only consistent factor related to both immediate and delayed pneumothorax. However, lobar location of the lesion was the most powerful variable in each group. The second most powerful variable was emphysema in affected lobe in immediate pneumothorax and the number of pleural punctures in delayed pneumothorax.

Lower lobe location is known to be a risk factor for pneumothorax owing to the greater mobility of lower lobes [25–28]. Significance of lesion location on delayed pneumothorax has not been demonstrated due to sparsity of related studies. A study by Choi et al. assessing risk factors for delayed pneumothorax failed to demonstrate the significance of lesion location as a contributing factor [17]. However, 10 of 15 cases of delayed pneumothorax occurred in upper lobes, while lesion distribution in the total population showed almost equal distributions (upper vs. middle/lower = 242 vs. 216). Mills et al. reported that the left upper lobe location was an independent risk factor for pneumothorax, in contrast to other studies, with delayed pneumothorax contributing to about one-third (66/253, 29.6%) of total pneumothorax in their study group [21]. Traill et al. reported their experience with two cases of pneumothorax occurring 26 and 36 hours after the procedure, in which each target lesion was located in the left and the right upper lobes, respectively [13]. Therefore, lesion location as a risk factor for delayed pneumothorax may differ from that in previous studies in which all cases of pneumothorax were included regardless of the onset time. Pleural injury involving lower lobes with higher aeration and ventilatory movement may lead to early appearance of pneumothorax. Conversely, pulmonary air in the upper lobes with less movement may escape slowly, resulting in late appearance of pneumothorax.

The risk of pneumothorax increases with increasing number of pleural punctures [29–31]. In the present study, the number of pleural punctures was related only to delayed pneumothorax. Conversely, emphysema was an independent variable related only to immediate pneumothorax; and the results were consistent with previous studies [17, 19]. Choi et al. found a significantly higher risk of immediate pneumothorax in patients with emphysema [17]. In contrast, the absence of an emphysema correlated with an increased rate of delayed pneumothorax. The disruption of dilated air spaces and the lack of elastic recoil in emphysematous lung may prevent rapid sealing of the air leak, resulting in early manifestations of pneumothorax [32, 33]. Meanwhile, the elastic recoil of the normal lung parenchyma and pleura over the lesion may seal the

small opening of the pleura initially to prevent pneumothorax [17]. Subsequent weakening of elastic recoil or multiple openings in normal pleura by multiple punctures may facilitate late presentation of pneumothorax. Therefore, it is speculated that pneumothorax appears immediately or later in CTLB depending on the speed of air leakage, which is affected by intactness of elastic recoil, the severity of injury, and expansibility of the targeted lung and overlying pleura.

In the present study, delayed pneumothorax showed higher requirement of chest tube drainage, than immediate pneumothorax (19.2% vs. 6.7%). Similar results were reported in previous studies regarding delayed pneumothorax [17, 20, 21]. However, since immediate and delayed pneumothorax were detected using different modalities, chest tube insertion rates between the two groups should not be directly compared. More than half of immediate pneumothorax was resolved quickly. Only 40.7% of immediate pneumothorax remained in initial follow-up chest radiographs and 12.7% of those cases required chest tube placement. Noh et al. reported similar results in which only 38% of CT detected pneumothorax showed persistence in follow-up chest radiographs at 4 h after biopsy and 21% of them required chest tube drainage [20]. Such results imply that a significant number of cases with immediate pneumothorax represent transient air leak via needle insertion site and resolve quickly when small pleural blood clots formed. In the contrary, immediate pneumothorax that is persistent on follow-up chest radiograph and delayed pneumothorax may represent continuous air leak, thus often requiring chest tube drainage. Therefore, clinical significance of delayed pneumothorax may be similar to that of immediate pneumothorax that is shown on initial follow-up chest radiograph obtained 4 h after procedure.

Our study has several limitations. First, the study was based on single institution and population of non-immediate cases were relatively small. Further studies with larger populations are required to corroborate our results. Second, due to retrospective study design, there may have been unidentified bias. Third, immediate and delayed pneumothorax was detected in CT and chest radiographs, respectively. We did not routinely acquire chest radiographs earlier than 4 h after procedure. However, this is common practice pattern in CTLB since it is not practical to obtain chest radiographs immediately after biopsy CT [20, 34]. Fourth, immediate and non-immediate pneumothorax was considered mutually exclusive but both events possibly occurred in a same patient. Therefore, each pneumothorax group was compared with the non-pneumothorax group.

In conclusion, pneumothorax tends to occur immediately after CTLB in patients with emphysema, lower lobe lesion, and long intrapulmonary biopsy track. Further attention and warnings are needed for those with multiple punctures of small lesions involving upper lobes due to the possibility of late development of pneumothorax and higher requirement for chest tube drainage.

## Supporting information

**S1 Table. Multivariate analysis of risk factors for delayed pneumothorax using immediate pneumothorax group as the reference.**
(DOCX)

## Author Contributions

**Conceptualization:** Kyungsoo Bae, Kyung Nyeo Jeon.

**Data curation:** Kyungsoo Bae, Ji Young Ha, Kyung Nyeo Jeon.

**Formal analysis:** Kyungsoo Bae.

**Investigation:** Kyungsoo Bae, Kyung Nyeo Jeon.

**Methodology:** Kyungsoo Bae, Ji Young Ha, Kyung Nyeo Jeon.

**Project administration:** Kyungsoo Bae, Ji Young Ha, Kyung Nyeo Jeon.

**Supervision:** Kyung Nyeo Jeon.

**Validation:** Kyungsoo Bae.

**Visualization:** Ji Young Ha.

**Writing – original draft:** Kyungsoo Bae, Kyung Nyeo Jeon.

**Writing – review & editing:** Kyungsoo Bae, Ji Young Ha, Kyung Nyeo Jeon.

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
