## [Decision Letter · Decision Letter 0]

15 Jul 2020

PONE-D-20-19268

Pneumothorax after CT-guided transthoracic lung biopsy: A comparison between immediate and delayed occurrence

PLOS ONE

Dear Dr. Jeon,

Thank you for submitting your manuscript to PLOS ONE. After careful consideration, we feel that it has merit but does not fully meet PLOS ONE’s publication criteria as it currently stands. Therefore, we invite you to submit a revised version of the manuscript that addresses the points raised during the review process.

ACADEMIC EDITOR: I have received the comments of the reviewers on your manuscript. The specific comments of the reviewers are included below. Please provide point by point response in your revised manuscript.

Please submit your revised manuscript by due date. If you will need more time than this to complete your revisions, please reply to this message or contact the journal office at plosone@plos.org. Please include the following items when submitting your revised manuscript:

We look forward to receiving your revised manuscript.

Kind regards,

Muhammad Adrish

Academic Editor

PLOS ONE

Journal Requirements:

2.  Please amend your Data availability statement to clarify how other researchers may obtain the datasets used in this study. We note for instance that no data has been provided in the Supplemental information.

3. We noticed you have some minor occurrence of overlapping text with previous publications, which needs to be addressed. In your revision ensure you cite all your sources (including your own works), and quote or rephrase any duplicated text outside the methods section. Further consideration is dependent on these concerns being addressed.

4. Thank you for stating the following financial disclosure: "NO"

Reviewers' comments:

Reviewer's Responses to Questions

**Comments to the Author**

1. Is the manuscript technically sound, and do the data support the conclusions?

Reviewer #1: Yes

Reviewer #2: Yes

2. Has the statistical analysis been performed appropriately and rigorously? 

Reviewer #1: Yes

Reviewer #2: No

3. Have the authors made all data underlying the findings in their manuscript fully available?

Reviewer #1: Yes

Reviewer #2: Yes

4. Is the manuscript presented in an intelligible fashion and written in standard English?

Reviewer #1: Yes

Reviewer #2: Yes

5. Review Comments to the Author

Reviewer #1: I read with intrest the manuscript.

Authors wrote a very good article. Congrats

Below only some suggestions:

1. introduction: wello wrote

2. Methods and results: are clear

3. Discussion: I appreciate the discussion , if you can improve with these two suggestions:

3a. compare with other data of literature

3b. the possible role of ultrasound. There are interesting use of ultrasound in thoracic diseases and dignosis (cite : Di Gennaro F, et al. Potential Diagnostic Properties of Chest Ultrasound in Thoracic Tuberculosis-A Systematic Review. Int J Environ Res Public Health. 2018;15(10):2235. Published 2018 Oct;

Bobbio F, et al. Focused ultrasound to diagnose HIV-associated tuberculosis (FASH) in the extremely resource-limited setting of South Sudan: a cross-sectional study. BMJ Open. 2019;9(4):e027179. Published 2019 Apr 2.)

Discuss the possible role of Ultrasound also as future persepctive.

Reviewer #2: In the present manuscript, the authors compared the risk factors for immediate and delayed pneumothorax after CT guided lung biopsy.

Major comments

The authors have not mentioned the reference category used for estimating the odds ratio of immediate and delayed pneumothorax. The immediate pneumothorax is mostly detected during the procedure and the diagnosis of delayed pneumothorax required subsequent X-ray. Therefore, it will be more useful if the authors look for the risk of delayed pneumothorax using the immediate pneumothorax as a reference category and the results may change.

Minor comments:

• 1.Statistics. The authors mentioned “The factors that were significant in the univariate analyses”. It should be a bivariate analysis.

• 2. Table1.

o It appears that the intrapulmonary biopsy track distances were not “normally distributed”. Hence, the use of the Student’s-t test for comparison was inappropriate.

o When the parameters were significantly different between total pneumothorax and without pneumothorax, the use of separate comparison of immediate and delayed pneumothorax and without pneumothorax is unnecessary duplication of results. It will be more useful to evaluate whether these parameters were significantly different between early and delayed pneumothorax cases.

o Do the distributions of gender and lesion type in groups were comparable?

Discussion

The authors mentioned “the elastic recoil of the normal lung parenchyma and

pleura over the lesion may seal the small opening of the pleura initially to prevent

pneumothorax” (page no 13). The elastic recoil of the lung and chest wall are in the opposite direction, how elastic recoil will prevent the pneumothorax.

6. PLOS authors have the option to publish the peer review history of their article (what does this mean?). If published, this will include your full peer review and any attached files.

Reviewer #1: **Yes: **Francesco Di Gennaro

Reviewer #2: No

---

## [Author Response · Author response to Decision Letter 0]

27 Jul 2020

Reviewer #1: 

I read with interest the manuscript.

Authors wrote a very good article. Congrats

Below only some suggestions:

1. introduction: well wrote

2. Methods and results: are clear

3. Discussion: I appreciate the discussion, if you can improve with these two suggestions:

3a. compare with other data of literature

 Thank you for your suggestion. 

There are a few articles regarding delayed pneumothorax in CT-guided biopsy, so we compared our results with that data. We highlighted the comparison in our manuscript.

3b. the possible role of ultrasound. 

 We added a short statement including the role of USG in lung biopsy in Introduction. We also added the article you mentioned to our reference list. 

There are interesting use of ultrasound in thoracic diseases and diagnosis (cite : Di Gennaro F, et al. Potential Diagnostic Properties of Chest Ultrasound in Thoracic Tuberculosis-A Systematic Review. Int J Environ Res Public Health. 2018;15(10):2235. Published 2018 Oct;

Bobbio F, et al. Focused ultrasound to diagnose HIV-associated tuberculosis (FASH) in the extremely resource-limited setting of South Sudan: a cross-sectional study. BMJ Open. 2019;9(4):e027179. Published 2019 Apr 2.)

Discuss the possible role of Ultrasound also as future persepctive.

Thank you very much for your insightful comments and suggestions which helped us improve the quality of our manuscript significantly.

 

Reviewer #2: 

In the present manuscript, the authors compared the risk factors for immediate and delayed pneumothorax after CT guided lung biopsy.

1) Major comments

The authors have not mentioned the reference category used for estimating the odds ratio of immediate and delayed pneumothorax. The immediate pneumothorax is mostly detected during the procedure and the diagnosis of delayed pneumothorax required subsequent X-ray. Therefore, it will be more useful if the authors look for the risk of delayed pneumothorax using the immediate pneumothorax as a reference category and the results may change.

 Thank you for your suggestion. 

For estimating the odds ratio of immediate and delayed pneumothorax, we used “no pneumothorax group” as our reference. The reason we used “no pneumothorax group” instead of “immediate pneumothorax group” was mentioned in limitations of the study in Discussion section.

Immediate and delayed pneumothorax are considered mutually exclusive. However, we thought both events could possibly occur in a same patient in reality. For example, when a patient had the pleura punctured twice or more, one puncture site could cause immediate pneumothorax and another puncture site could cause delayed pneumothorax. (However, we cannot discriminate the two types of pneumothorax. Therefore, such case may be considered as immediate pneumothorax which persists on follow-up chest X-rays). Another example is when immediate pneumothorax develops, resolves soon, then develops again (delayed pneumothorax). However, we may also consider such case as immediate pneumothorax which is persistent on follow-up X-rays.

We also had the same question you mentioned. Therefore, we looked for the risk factors of delayed pneumothorax using the immediate pneumothorax as a reference. Lobar location of the lesion and No. of pleural punctures were revealed as risk factors of delayed pneumothorax. Lesion size did not show statistical significance. When immediate pneumothorax group was used as the reference, the association between delayed pneumothorax and lobar location of lesion was shown to be greater. 

Multivariate analysis of risk factors for delayed pneumothorax 

Middle/lower lobe: upper lobe 0.074 (CI = 0.022-0.248)*

No. of pleural punctures 2.672 (CI = 1.477-4.837)*

* Immediate pneumothorax was used as the reference group.

Data are presented as odds ratios (95% confidence interval [CI]).

We will provide this result in the Supplemental information. 

Minor comments:

1.Statistics. The authors mentioned “The factors that were significant in the univariate analyses”. It should be a bivariate analysis. 

 Thank you for your suggestion. We agree with the reviewer. Therefore, we corrected the term ‘univariate’ to ‘bivariate’. 

2. Table1.

o It appears that the intrapulmonary biopsy track distances were not “normally distributed”. Hence, the use of the Student’s-t test for comparison was inappropriate.

 We agree with the reviewer. Since the length of intrapulmonary biopsy track distances did not show normal distribution, we recalculated using Mann-Whitney U test. The differences between groups were significant (except between immediate and delayed pneumothorax, which was not included in our manuscript).

No PNX (n=375) vs. Total PNX (n=161), p <0.001

No PNX (n=375) vs. Immediate PNX (n=135), p = 0.005

No PNX (n=375) vs. Delayed PNX (n=26), p = 0.004

(Immediate PNX (n=135) vs. Delayed PNX (n=26), p = 0.118)

* PNX = pneumothorax

3. When the parameters were significantly different between total pneumothorax and without pneumothorax, the use of separate comparison of immediate and delayed pneumothorax and without pneumothorax is unnecessary duplication of results. It will be more useful to evaluate whether these parameters were significantly different between early and delayed pneumothorax cases.

 Thank you for your suggestion. From our experience, we thought that several inconsistent risk factors for pneumothorax in previous literatures are, at least partly, owing to not considering the developing time of the pneumothorax. Therefore, we tried to see the risk factors for immediate and delayed pneumothorax separately. In statistics, we compared each pneumothorax group with no pneumothorax group for reasons we explained in your major comments. 

4. Do the distributions of gender and lesion type in groups were comparable?

 Thank you for your suggestion. We rechecked our data. There were no differences in the distributions of gender and lesion type between groups.

5. Discussion

The authors mentioned “the elastic recoil of the normal lung parenchyma and

pleura over the lesion may seal the small opening of the pleura initially to prevent pneumothorax” (page no 13). The elastic recoil of the lung and chest wall are in the opposite direction, how elastic recoil will prevent the pneumothorax.

 We agree with the reviewer that the elastic recoils of the lung and chest wall act in the opposite direction. However, we meant elastic recoil of “the lung and visceral pleura”. Intact elastic recoil makes punctured pleural hole become smaller. We added a reference for the description.

Thank you very much for your insightful comments and suggestions which helped us improve the quality of our manuscript significantly

---

## [Decision Letter · Decision Letter 1]

11 Aug 2020

Pneumothorax after CT-guided transthoracic lung biopsy: A comparison between immediate and delayed occurrence

PONE-D-20-19268R1

Dear Dr. Jeon,

We’re pleased to inform you that your manuscript has been judged scientifically suitable for publication and will be formally accepted for publication once it meets all outstanding technical requirements.

Kind regards,

Muhammad Adrish

Academic Editor

PLOS ONE

Additional Editor Comments (optional):

Reviewers' comments:

Reviewer's Responses to Questions

**Comments to the Author**

1. If the authors have adequately addressed your comments raised in a previous round of review and you feel that this manuscript is now acceptable for publication, you may indicate that here to bypass the “Comments to the Author” section, enter your conflict of interest statement in the “Confidential to Editor” section, and submit your "Accept" recommendation.

Reviewer #1: All comments have been addressed

Reviewer #2: All comments have been addressed

2. Is the manuscript technically sound, and do the data support the conclusions?

Reviewer #1: Yes

Reviewer #2: Yes

3. Has the statistical analysis been performed appropriately and rigorously? 

Reviewer #1: Yes

Reviewer #2: Yes

4. Have the authors made all data underlying the findings in their manuscript fully available?

Reviewer #1: Yes

Reviewer #2: Yes

5. Is the manuscript presented in an intelligible fashion and written in standard English?

Reviewer #1: Yes

Reviewer #2: Yes

6. Review Comments to the Author

Reviewer #1: I appreciate your manuscript

Authors improved their manuscript

I think can be useful to scientific community

Reviewer #2: (No Response)

7. PLOS authors have the option to publish the peer review history of their article (what does this mean?). If published, this will include your full peer review and any attached files.

Reviewer #1: **Yes: **Francesco Di Gennaro

Reviewer #2: No

---

## [Editor Report · Acceptance letter]

13 Aug 2020

PONE-D-20-19268R1 

Pneumothorax after CT-guided transthoracic lung biopsy: A comparison between immediate and delayed occurrence 

Dear Dr. Jeon:

I'm pleased to inform you that your manuscript has been deemed suitable for publication in PLOS ONE. Congratulations! Your manuscript is now with our production department. 

Kind regards, 

on behalf of

Dr. Muhammad Adrish 

Academic Editor

PLOS ONE